# Free-Space Transmission and Detection of Variously Polarized Near-IR Beams Using Standard Communication Systems with Embedded Singular Phase Structures

**DOI:** 10.3390/s22030890

**Published:** 2022-01-24

**Authors:** Sergey V. Karpeev, Vladimir V. Podlipnov, Svetlana N. Khonina, Nikolay A. Ivliev, Sofia V. Ganchevskay

**Affiliations:** 1IPSI RAS-Branch of the FSRC “Crystallography and Photonics” RAS, 443001 Samara, Russia; karp@ipsiras.ru (S.V.K.); khonina@ipsiras.ru (S.N.K.); ivlievn@gmail.com (N.A.I.); son7755@yandex.ru (S.V.G.); 2Samara National Research University, 443086 Samara, Russia

**Keywords:** free-space optics, polarization recognition, efficient beam coupling, blazed grating

## Abstract

We propose to achieve multichannel information transmission in free space by means of variously polarized beams. The interaction of vortex beams of various orders with the main polarization states is theoretically analyzed. The passage of beams with different polarization states through multi-order diffractive optical elements (DOEs) is simulated numerically. Using the simulation results, tables of code correspondence of diffraction order numbers to the presence of phase vortices in the analyzed beams are constructed, which allow one to determine diffraction orders that carry information about various polarization states. The performed experiment made it possible to study the recognition of the first order cylindrical polarization state formed by a Q-plate converter using a phase DOE. In the experiment, these elements were built into a commercial fiber-optic communication system operating at the near-IR frequencies. After detecting the beam polarization state, beams of the required diffraction orders are efficiently coupled into optical fiber using an additional phase element. The developed optical detection system also provides channel suppression of homogeneously polarized components, which are supposed to be used for transmission of other channels.

## 1. Introduction

Multichannel data transmission in free space can be achieved by means of laser beams with different properties [1,2]. The most common approach is the use of vortex beams of various orders [3,4,5,6,7], as well as cylindrical vector beams [8,9]. To multiplex information transmission channels, it is also possible to exploit both phase and polarization degrees of freedom [10,11,12]. An integral part of the implementation of optical communication is the multiplexing of the transmitted signal and the demultiplexing of the received signal on the receiving side, which in both cases rely on the use of special optical elements based on diffraction gratings [13,14,15,16,17].

The possibility of multichannel (simultaneous) analysis of vortex states of laser beams for homogeneous types of polarization was first shown in [18]. This approach makes it possible to measure even the fractional orbital angular momentum of the beam [19] and is successfully used for (de-)multiplexing vortex beams [20,21,22,23].

The use of beam polarization as an additional degree of freedom complicates the task of demultiplexing. As a rule, the polarization state is identified using polarizing optics elements. This requires a change in the position of the polarizing element during the analysis, which significantly slows down the research and reduces its efficiency. However, in a number of cases, for example, for certain types of polarizations, it is possible to use non-polarizing optics, for example, singular optics elements. For the first time, the idea of using fork-shaped polarization phase grating for detecting radially and azimuthally polarized first-order beams was proposed in [24]. The polarization selectivity used by the phase grating was based on the employment of a liquid crystal display and additional two quarter-wave plates. The performed experiments demonstrated a significant difference between the results for circular and nonhomogeneous polarizations, as well as the impossibility of distinguishing between radial and azimuthal polarizations.

The use of vortex phase elements in combination with tight focusing and high-aperture diffractive axicons [25] makes it possible to distinguish between certain types of polarizations (in particular, to distinguish radial and azimuthal first-order polarizations) and optical vortices of the first and second orders. Deng et al. [26] considered the detection of vortex beams with linear *x*- and *y*-polarizations based on the selective transmission of SLMs (the *x*-component of the radiation is transmitted, and the *y*-component is reflected). Finally, Khonina et al. [27] proposed to use multi-order DOEs for a complex analysis of beams, including the determination of the polarization-phase state for certain sets of orders, with the formation of a digital code that uniquely determines the order of the vortex singularity and the order of cylindrical polarization of the beam in various combinations.

Note, in the paraxial regime, it is impossible to distinguish the type of polarization (radial or azimuthal). The use of polarizing optical devices only partially solves the problem of identifying the polarization state, and rotation of a polarization analyzer is often required. Therefore, in this work, we consider additional diffractive orders used to reveal phase relationships between main diffractive orders, and to distinguish between radial and azimuthal polarizations using only one stationary polarization analyzer.

All of the above methods use the intensity distribution in diffraction orders as an indicator of the presence of certain beams, and in the case of identifying the polarization state of the beam, a simultaneous analysis of several diffraction orders is required. On the other hand, to transmit signals in the atmosphere, it is convenient to use existing commercial fiber-optic communication systems with replacement of fiber sections with free space and modification of a number of system elements [28,29].

As is known, free space optical (FSO) communication systems operate in the paraxial regime; therefore, to detect the polarization states of beams, it is necessary to combine all the necessary diffraction orders into a parallel beam with further focusing by the collimator lens on the end face of the receiving fiber. Thus, it is proposed, after the multichannel DOE, which forms the required diffraction orders, to add a phase element that selects the required diffraction orders and compensates for the tilt of the beams of the selected diffraction orders. The possibilities of diffractive optics make it possible to realize such a phase element in the form of a single DOE.

Thus, to create information transmission systems with polarization-division multiplexing [9,10,30], it is necessary, on the one hand, to develop methods for detecting polarization states of beams using non-polarizing elements, namely, multi-order DOEs. On the other hand, optical systems must be built to separate and transform information diffraction orders with the aim of their further input into a communication system based on the use of optical fibers. These issues are discussed in Section 2 and Section 3 of our work.

## 2. Theory and Simulation

Let us consider the interaction of vortex beams of different orders with the main polarization states.

### 2.1. Homogeneous Types of Polarization

For a vortex field with a homogeneous type of polarization (linear, circular, or elliptical), we can write:(1)$$F_{m}(r,\phi )=A(r)\exp\left(im\phi \right)e_{h}=A(r)\exp\left(im\phi \right)\left(\begin{array}{c} c_{x}\\ c_{y} \end{array}\right)$$
where *m* is the order of the optical vortex, A(r) is a radius-dependent arbitrary function, and $e_{h}={\left(c_{x},c_{y}\right)}^{T}$ is the vector of the homogeneous polarization state.

The vortex field (1) has many orthogonal states in accordance with the order of the vortex phase singularity *m*, which can be unambiguously detected using multi-order DOEs [17,18,19,20,21,22,23,31,32,33]; although, the polarization state is not recognized.

The complex transmission function of a multi-order DOE, matched with a set of optical vortices, has the form:(2)$$\tau (x,y)=\sum_{q=-Q}^{Q}\exp\left(iq\phi \right)\exp\left(i\alpha_{q}x+i\beta_{q}y\right)$$
where *q* is the diffraction order number, which in this case coincides with the topological charge of the optical vortex; and α*_q_* and β*_q_* are the spatial carrier frequencies of the corresponding diffraction order. Note that DOE (2) can be made binary if the spatial frequencies for the complex conjugated vortex orders are selected equal to $\alpha_{-q}=-\alpha_{q},\;\beta_{-q}=-\beta_{q}$ [23].

Let DOE (2) be supplemented with a spherical lens and illuminated with a light beam (1). The distribution in the focal plane is described by the Fourier transform:(3)$$G(u,v)\approx -i\frac{k}{f}\sum_{q=-Q}^{Q}C_{q}\delta \left(u-\frac{\lambda f}{2\pi }\alpha_{q},v-\frac{\lambda f}{2\pi }\beta_{q}\right),$$
where $\delta \left(u-u_{q},v-v_{q}\right)$ are the shifted delta functions, and $C_{q}$ are the vector expansion coefficients for a homogeneously polarized field:(4)$$C_{q}^{h}=\left(\begin{array}{c} c_{x}\\ c_{y} \end{array}\right)\delta_{m,q}\cdot I_{R}$$
where $I_{R}=\int_{0}^{R}A(r)r\;dr$, and $\delta_{s,q}=\{\begin{array}{c} 1,\;s=q,\\ 0,\;s\neq q. \end{array}$.

Detection is carried out by the squared moduli of the coefficients at the centers of the corresponding diffraction orders:(5)$${\left|C_{q}^{h}\right|}^{2}=\delta_{m,q}\cdot {\left|I_{R}\right|}^{2}\left({\left|c_{x}\right|}^{2}+{\left|c_{y}\right|}^{2}\right)=\delta_{m,q}\cdot {\left|I_{R}\right|}^{2}$$

As follows from expression (5), the polarization state in the case under consideration does not in any way affect the squared moduli of the coefficients. Obviously, the squared moduli of the coefficients ${\left|C_{q}^{h}\right|}^{2}$ makes it impossible to distinguish the type of homogeneous polarization, and only the order of the phase singularity *m* can be determined, taking into account the sign.

The form of the phase function of an eight-channel DOE, matched with optical vortices, is shown in Figure 1a. The zero order (*m* = 0) is shifted from the optical axis in accordance with formula (2) by adding the carrier frequency. The central part of the focal plane is not used for useful information in order to reduce the influence of interferences localized in the central part.

To make the phase function of the optical element binary (convenient for fabrication), the zero vortex diffraction orders are doubled. The pattern formed in the focal plane when the DOE is illuminated by a Gaussian beam and the correspondence of diffraction orders are shown in Figure 1b. The action of such an element in the analysis of vortex beams with a homogeneous type of polarization is shown in Figure 2.

The code correspondence of the numbers of diffraction orders to the presence of phase vortices in the analyzed beam for a homogeneous polarization state is presented in Table 1. As can be seen, the code sequences for individual vortex orders are orthogonal, which allows for unambiguous identification.

As can be seen from the results presented in Figure 2 and in Table 1, the multichannel DOE makes it possible to unambiguously detect the presence of optical vortices of various orders (separate or superposition) in a homogeneously polarized beam. If there are several phase vortices, then, in fact, the patterns corresponding to individual vortices are added. This fact is in full agreement with expression (3).

An increase in the number of detected orders requires an increase in the diffraction orders with which the DOE is matched [20,21].

### 2.2. Cylindrical Types of Polarization

Next, we consider the action of DOE (2) in the analysis of vortex cylindrically polarized beams.

A vortex field with cylindrical polarization of the *p*th order (radial, azimuthal, or mixed) is expressed as:(6)$$F_{p,m}(r,\phi )=A(r)\exp\left(im\phi \right)e_{p}=A(r)\exp\left(im\phi \right)\left(\begin{array}{c} \cos\left(p\phi +\phi_{0}\right)\\ \sin\left(p\phi +\phi_{0}\right) \end{array}\right)$$
where $\phi_{0}$ is the phase shift ($\phi_{0}$ = 0 corresponds to radial polarization, and $\phi_{0}$ = π, to azimuthal polarization), and $e_{p}$ is the vector of the cylindrical polarization state of the *p*th order.

Obviously, field (6) has two degrees of freedom—the vortex singularity order *m* and the cylindrical polarization order *p*. In this case, the vector expansion coefficients have the following forms:(7)$$C_{q}^{Rd\_p}=\frac{1}{2}\left\{\left(\begin{array}{c} 1\\ -i \end{array}\right)\delta_{m+p,q}+\left(\begin{array}{c} 1\\ i \end{array}\right)\delta_{m-p,q}\right\}\cdot I_{R}$$
for a radially polarized field and
(8)$$C_{q}^{Az\_p}=\frac{1}{2}\left\{\left(\begin{array}{c} i\\ 1 \end{array}\right)\delta_{m+p,q}+\left(\begin{array}{c} -i\\ 1 \end{array}\right)\delta_{m-p,q}\right\}\cdot I_{R}$$
for an azimuthally polarized field.

It follows from expressions (4), (7), and (8) that, in contrast to homogeneous polarization (1), a cylindrically polarized field leads to the formation of two (rather than one) correlation peaks corresponding to the sum and difference in the orders of the vortex singularity and polarization: s=m±p.

For cylindrical vector beams, it is also impossible to distinguish between the type of polarization (radial or azimuthal), since ${\left|C_{q}^{Rd\_p}\right|}^{2}={\left|C_{q}^{Az\_p}\right|}^{2}$; however, it is possible to determine both the order of polarization *p* and the magnitude of the phase singularity *m* [27].

The action of DOE (2) in the analysis of vortex beams with a cylindrical type of polarization is shown in Figure 3 and Figure 4, and the code correspondence of the numbers of diffraction orders to the order of polarization and the presence of phase vortices in the analyzed beam is presented in Table 2 and Table 3.

As can be seen from the results presented in Figure 3 and Figure 4, in the presence of separate phase vortices in a cylindrically polarized beam, a pair of correlation orders is formed, from the location of which one can determine both the polarization order *p* and the value of the phase singularity *m* (the code key is described in more detail in [27]).

Note that the code sequences for individual vortex orders are partially orthogonal, taking into account the polarization order (Table 2); therefore, unambiguous identification is carried out only in the presence of certain code sequences, which protects information from unauthorized decryption. With an increase in the polarization order *p*, it is necessary to increase the number of diffraction orders; otherwise, identification is carried out for smaller orders of vortices (Table 3).

While in the case of cylindrical polarization the number of degrees of freedom for the multiplexing of information transmission channels is significantly increased due to the polarization order, it is desirable to be able to distinguish between different polarization states.

Note that the use of polarizing optical devices only partially solves the problem of identifying the polarization state. In particular, even when separating the intensity of individual components of the electric field (see Table 4), the intensity of diffraction orders makes it problematic to distinguish diagonal linear polarizations, circular polarizations of different directions, as well as radial and azimuthal ones, since the main difference is contained in the phase information.

As one can see, it is more difficult to distinguish homogeneous polarizations than cylindrical ones. The peculiarity of cylindrical polarization is associated with the presence of pair orders (8) and (9), which have different polarization states (circular polarization of the opposite direction). In this paper, we propose to strengthen these differences by addition (or superposition) of the expansion functions. This approach is used to reconstruct the phase of the wavefront [34,35,36].

### 2.3. Theoretical Description of the Method for Detecting the Polarization State Based on the Superposition of Orders

To obtain additional information about the polarization state of cylindrical beams, one can use superpositions of vortices, in particular, diffraction orders matched with the functions cos(qϕ) and sin(qϕ).

Figure 5 shows the form and operation of a 9-channel DOE, which, in addition to optical vortices of the first and second order, is also consistent with their superpositions. Note that such superpositions can be realized by the interference of vortex orders. This can be achieved by reducing various diffraction orders with additional prisms.

The action of the diffraction orders matched with cos(qϕ) and sin(qϕ) has a different effect for different types of cylindrical polarization.

For a radially polarized field, we have:$$\begin{array}{l} C_{q}^{Rd\_p}=\frac{1}{2}\int_{0}^{2\pi }\int_{0}^{R}\exp\left(im\phi \right)\left(\begin{array}{c} \cos\left(p\phi \right)\\ \sin\left(p\phi \right) \end{array}\right)\sin(q\phi )d\phi \;r\;dr\;\;=\\ =\frac{1}{4}\int_{0}^{2\pi }\int_{0}^{R}\exp\left(im\phi \right)\left(\begin{array}{c} \sin\left[\left(q-p\right)\phi \right]+\sin\left[\left(q+p\right)\phi \right]\\ \cos\left[\left(q-p\right)\phi \right]-\cos\left[\left(q+p\right)\phi \right] \end{array}\right)d\phi \;r\;dr\;\;=\\ =\frac{1}{4}\mathrm{sgn}(m)\left\{\left(\begin{array}{c} i\\ -1 \end{array}\right)\delta_{\left|m\right|,q-p}+\left(\begin{array}{c} i\\ 1 \end{array}\right)\delta_{\left|m\right|,q+p}\right\}\cdot I_{R} \end{array}$$
$$\begin{array}{l} C_{q}^{Rd\_p}=\frac{1}{2}\int_{0}^{2\pi }\int_{0}^{R}\exp\left(im\phi \right)\left(\begin{array}{c} \cos\left(p\phi \right)\\ \sin\left(p\phi \right) \end{array}\right)\cos(q\phi )d\phi \;r\;dr\;\;=\\ =\frac{1}{4}\int_{0}^{2\pi }\int_{0}^{R}\exp\left(im\phi \right)\left(\begin{array}{c} \cos\left[\left(q-p\right)\phi \right]-\cos\left[\left(q+p\right)\phi \right]\\ -\sin\left[\left(q-p\right)\phi \right]+\sin\left[\left(q+p\right)\phi \right] \end{array}\right)d\phi \;r\;dr\;\;=\\ =\frac{1}{4}\mathrm{sgn}(m)\left\{\left(\begin{array}{c} -1\\ -i \end{array}\right)\delta_{\left|m\right|,q-p}+\left(\begin{array}{c} 1\\ i \end{array}\right)\delta_{\left|m\right|,q+p}\right\}\cdot I_{R} \end{array}$$

For an azimuthally polarized field, we have:$$\begin{array}{l} C_{q}^{Az\_p}=\frac{1}{2}\int_{0}^{2\pi }\int_{0}^{R}\exp\left(im\phi \right)\left(\begin{array}{c} -\sin\left(p\phi \right)\\ \cos\left(p\phi \right) \end{array}\right)\sin(q\phi )d\phi \;r\;dr\;\;=\\ =\frac{1}{4}\int_{0}^{2\pi }\int_{0}^{R}\exp\left(im\phi \right)\left(\begin{array}{c} -\cos\left[\left(q-p\right)\phi \right]+\cos\left[\left(q+p\right)\phi \right]\\ \sin\left[\left(q-p\right)\phi \right]+\sin\left[\left(q+p\right)\phi \right] \end{array}\right)d\phi \;r\;dr\;\;=\\ =\frac{1}{4}\mathrm{sgn}(m)\left\{\left(\begin{array}{c} 1\\ i \end{array}\right)\delta_{\left|m\right|,q-p}+\left(\begin{array}{c} -1\\ i \end{array}\right)\delta_{\left|m\right|,q+p}\right\}\cdot I_{R} \end{array}$$
$$\begin{array}{l} C_{q}^{Az\_p}=\frac{1}{2}\int_{0}^{2\pi }\int_{0}^{R}\exp\left(im\phi \right)\left(\begin{array}{c} -\sin\left(p\phi \right)\\ \cos\left(p\phi \right) \end{array}\right)\cos(q\phi )d\phi \;r\;dr\;\;=\\ =\frac{1}{4}\int_{0}^{2\pi }\int_{0}^{R}\exp\left(im\phi \right)\left(\begin{array}{c} -\sin\left[\left(q-p\right)\phi \right]-\sin\left[\left(q+p\right)\phi \right]\\ \cos\left[\left(q-p\right)\phi \right]+\cos\left[\left(q+p\right)\phi \right] \end{array}\right)d\phi \;r\;dr\;\;=\\ =\frac{1}{4}\mathrm{sgn}(m)\left\{\left(\begin{array}{c} -i\\ -1 \end{array}\right)\delta_{\left|m\right|,q-p}+\left(\begin{array}{c} -i\\ -1 \end{array}\right)\delta_{\left|m\right|,q+p}\right\}\cdot I_{R} \end{array}$$

As follows from the above expressions, both in-phase and anti-phase additions will occur in different components of the field, depending on the type of polarization, which will allow them to be distinguished. This is shown more clearly in Table 5 and Table 6, where the circles display the diffraction orders, which allow, in the presence of a stationary polarization analyzer (without rotation), the type of polarization (radial or azimuthal) to be unambiguously identified. Note that in this case the presence of correlation peaks is not necessary, since the diffraction orders marked with circles differ significantly in total energy for different types of polarization.

Thus, additional orders used for the interference of vortex orders make it possible to reveal phase relationships between pair orders and to distinguish between radial and azimuthal polarizations using only one additional stationary (without rotation) polarization analyzer.

In order to recognize the polarization state of a vortex beam without using polarizing optics elements, it is possible to use other superpositions, which can also be realized by reducing various diffraction orders by additional prisms.

The developed approach can be applied in information transmission systems with polarization-division multiplexing [9,10,30]. This will increase the number of possible polarization states used for multiplexing and coding communication channels.

## 3. Materials and Methods

The method for recognizing the polarization state of a beam is based on the use of a phase element in the form of a fork-shaped grating, which is matched with vortices of the order of *m* = ± 1. The performed simulation (see Figure 3a) showed that when a multi-order DOE is illuminated by a beam with a first-order cylindrical polarization state, two diffraction maxima are formed corresponding to vortices of the order of *m* = ± 1. For beams with homogeneous polarization that do not contain vortices, circular intensity distributions are observed in diffraction orders.

Figure 6 shows a schematic of the experimental setup that implements this method. The optical fiber shown in the figure is the output of a transceiver that temporarily modulates the beam to transmit information.

The plane-polarized beam emerging from the fiber is collimated by lens L1 and then passes through a variable spiral plate (VSP) (ARCoptics, Neuchatel, Switzerland), which forms a first-order cylindrical vector beam for a wavelength of 1.55 μm used in the transceiver. Depending on the orientation of the polarization axis, the beam can be both radially and azimuthally polarized. Then, after passing through a section of free space, the beam is coupled into the receiving part for decoding. The purpose of decoding is to identify the cylindrically polarized component of the beam and to launch it into the collimator of the optical fiber of the receiving part of the transceiver. Obviously, to identify the type of beam polarization by diffraction orders, it is necessary to separate the central part of the diffraction order using a point diaphragm of the corresponding diameter. The diameter of the central maximum of the diffraction order for vortices with *m* = ±1 is close to the size of the diffraction spot. This filtering of diffraction orders makes it possible to distinguish annular distributions in orders from diffraction maxima (zeros from ones in the correspondence table). In addition, the selection of the required orders makes it possible to suppress radiation with other types of polarization (if present), for example, in our case, plane-polarized radiation. The launching of radiation into fiber after filtration is hindered by the off-axis position of the diffraction orders, which leads to the tilt of the beam axes relative to the optical axis after passing through lens L3. To compensate for the tilt, it is proposed to use a blazed diffraction grating, which is an analogue of a conventional prism. To correct the tilt angles of propagation of beams of diffraction orders, the blazed grating should consist of two halves with mirror-symmetric sawtooth profiles with a period that is the same as that of the fork-shaped grating. The blazed grating forms ghost images of the diffraction orders on the optical axis where they are aligned. It is easy to see that the height of the grating teeth profile in this case must increase from the center to the periphery. The resulting beams are collimated by lens L3 and then can be efficiently coupled into optical fiber with a standard collimator lens L4.

The fork-shaped grating was made by etching a quartz substrate through a chromium mask deposited using a circular laser writing station CLWS-200s. A fragment of the profile measured with a KLATencor grating profilometer is shown in Figure 7, with the calculated relief height and the period being 1.7 μm and 120 μm, respectively.

Point diaphragms were made by thermochemical oxidation of thin chromium films on a CLWS-200 setup. Holes of different diameters are used to match the sizes of diffraction orders for different sizes of the input beam. The center hole is used for accurate alignment with the zero-order diffraction in the system. Further alignment of +1 and −1 orders is carried out by simple rotation of the substrate with the diaphragms. Then, the central hole is closed. The view of the diaphragm is shown in Figure 8.

The blazed grating was made by direct laser writing using a circular laser writing station CLWS-2014. The lithography process was carried out in a 6 μm thick FP-3535 positive photoresist layer, which was previously deposited on a quartz substrate by centrifugation. To obtain an exact correspondence of the height and slope of the blazed grating profile, additional studies were performed to select the lithography regime. At the final writing cycle, the deviation of the microrelief height did not exceed 100 nm. A 3D image of the obtained relief and grating profile with a height of 1.75 μm and a period of 120 μm, measured with a Zygo NewView 5000 optical interferometer, is shown in Figure 9.

As can be seen from the figure, the microrelief phase is quite linear, and a small defect area near the center, characteristic of a circular laser writing station, is outside the working zone of the grating. The teeth of the grating are also inclined in the right direction.

To obtain images of the intensity distribution formed in different parts of the optical system, we used a VS-320 IR camera (KB-Vita, Russia) equipped with an InGaAs sensor having a resolution of 320–256 pixels. The sensor has a pixel size of 30 μm and a spectral sensitivity of 0.9–1.7 μm.

## 4. Results of the Experiment

First, we tested and adjusted the Q-plate. Figure 10 shows the intensity distributions in the beams passed through the Q-plate at different positions of the polarizer axis.

One can see that the state of the beam polarization is close to the first-order radial polarization. Figure 11 shows the intensity distributions in the focal plane of lens L2 for plane polarization (in the absence of a Q-plate) and for azimuthal polarization (in the presence of a Q-plate).

Figure 11a demonstrates the diffraction orders of the ring, and Figure 11b shows the maxima in the centers of the orders. This testifies to the efficiency of the proposed approach. Further (see Figure 12), the intensity distributions at different distances are shown after passing through the blazed grating.

In Figure 12, distances both between the centers of the beams and the diameters differ for different distances. This shows the presence of the divergence of the beams themselves and the tilt of their axes. Analysis of the distance between the centers of the beams and their diameters shows that the axes intersect in the plane of the diaphragm (See Figure 6). Thus, there are ghost images of the diffraction orders on the optical axis.

Finally, Figure 13 shows the intensity distributions in the same diffraction orders after passing through lens L3 at different distances from the lens. Lens 3 serves both to compensate for the divergence of the beams themselves and to form two beams parallel to the optical axis.

The retention of the size of the intensity distributions indicates the absence of beam divergence. Beams parallel to the optical axis are focused by the collimator lens onto the optical axis, where the input end face of the optical fiber is located.

Thus, it was possible to achieve a high efficiency of coupling into the optical fiber, which allows for stable atmospheric communication under the same conditions and on the same equipment as in [28,29].

## 5. Conclusions

An optical system for decoding the polarization states of beams in commercial near-IR communication systems was studied theoretically and experimentally. Numerical simulation of the passage of beams with different polarization states through multi-order diffractive optical elements showed the possibility of recognizing the orders of vortex fields with homogeneous polarization, as well as cylindrical polarizations by the presence of pair orders. Based on the simulation results, we constructed tables of code correspondence of diffraction order numbers to the presence of phase vortices in the analyzed beams, which make it possible to determine the diffraction orders that carry information about various polarization states. Code correspondence tables allow one to develop and implement an optical system for detecting cylindrically polarized beams.

A phase element was developed and manufactured to separate the beams of the required orders with the correction of their propagation angles for efficient coupling into optical fiber while simultaneously suppressing homogeneously polarized components. An experiment was performed using a commercial fiber-optic data transmission system operating in the near-IR range, supplemented by two DOEs, one of which is a fork-shaped binary-phase grating, and the other is a segmented blazed grating that corrects the propagation directions of light beams. Using an electrically controlled Q-plate, a first-order cylindrical vector beam was produced, which, in the case of incidence on a fork-shaped grating, gives two maxima in ±1 diffraction orders. Then, using the blazed grating and an additional lens, the beams of diffraction orders are directed into a collimator, followed by coupling into fiber connected to a receiver. Other communication channels with polarization multiplexing can be transmitted by beams with homogeneous polarization, which are suppressed in this system for detecting a beam with nonhomogeneous polarization.

## Figures and Tables

**Figure 1 sensors-22-00890-f001:**
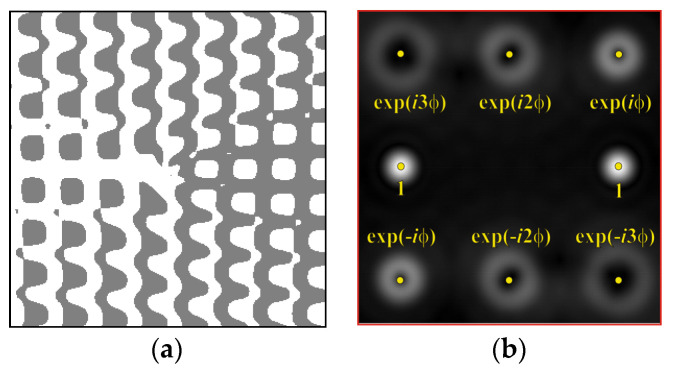
Eight-channel DOE, matched with optical vortices: (**a**) the form of the phase function, and (**b**) the correspondence of diffraction orders in the focal plane.

**Figure 2 sensors-22-00890-f002:**
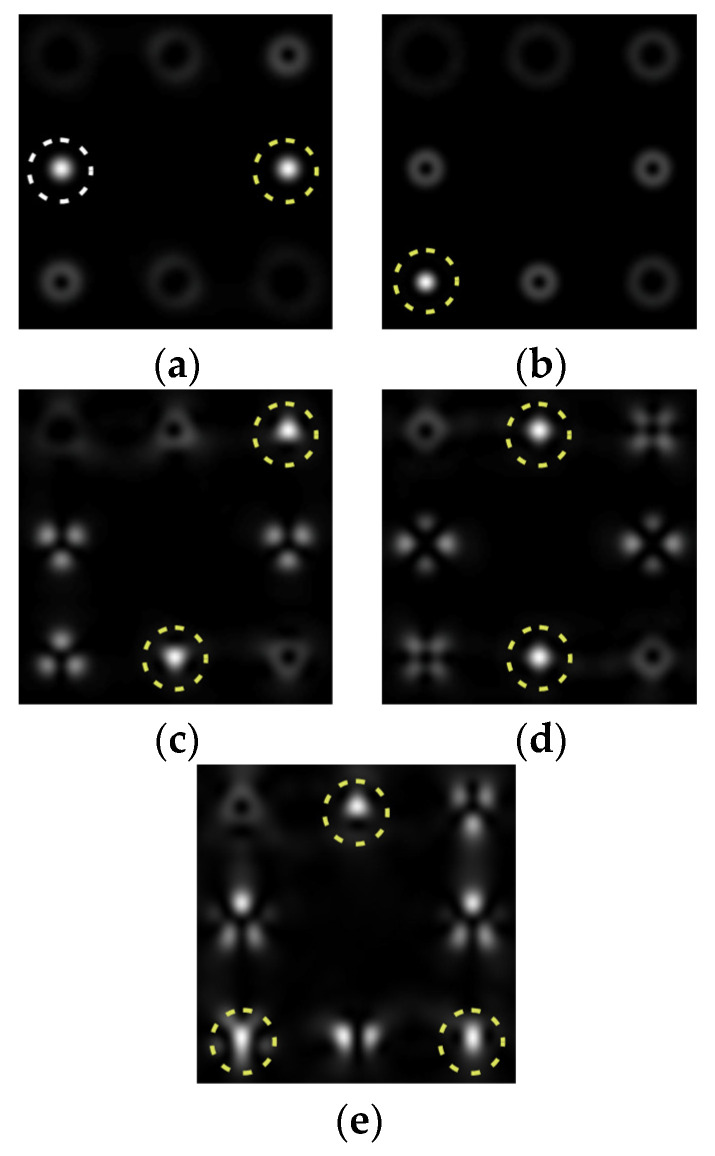
Action of an eight-channel DOE, matched with optical vortices in the analysis of vortex fields with homogeneous polarization: (**a**) *m* = 0, (**b**) *m* = −1, (**c**) *m* = 1, −2, (**d**) *m* = 2, −2, and (**e**) *m* = −1.2, −3.

**Figure 3 sensors-22-00890-f003:**
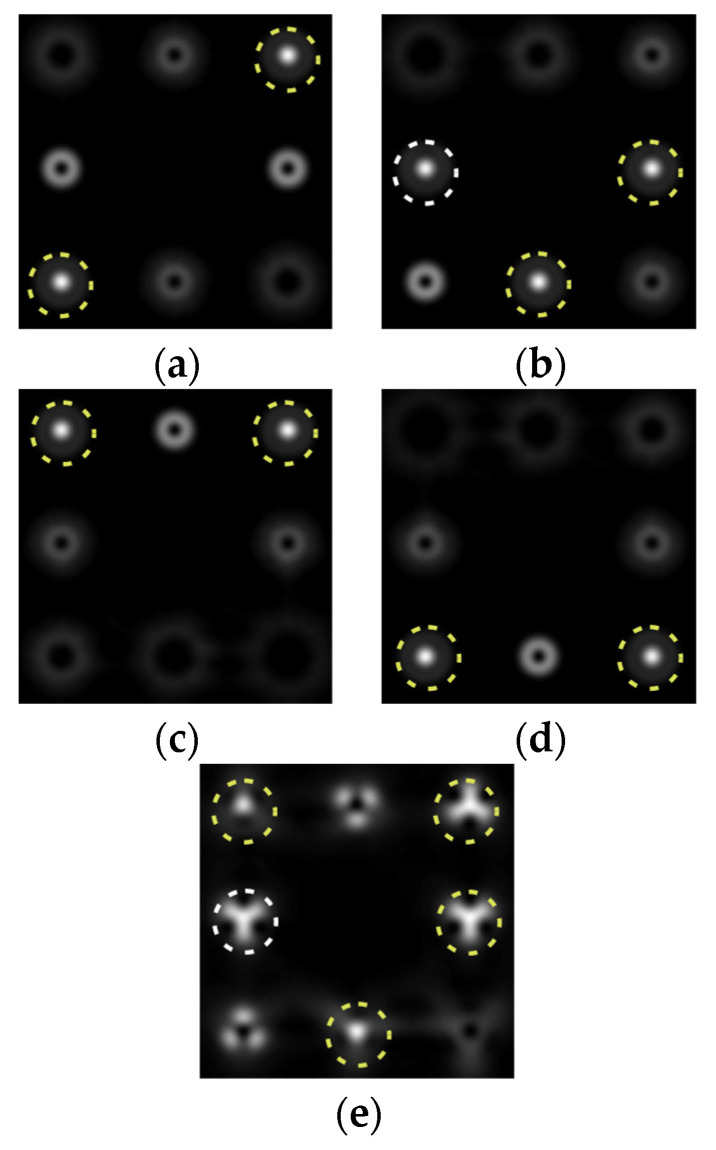
Action of an eight-channel DOE, matched with optical vortices in the analysis of vortex fields with cylindrical polarization of the first order (*p* = 1): (**a**) *m* = 0, (**b**) *m* = −1, (**c**) *m* = 2, (**d**) *m* = −2, and (**e**) *m* = −1, 2.

**Figure 4 sensors-22-00890-f004:**
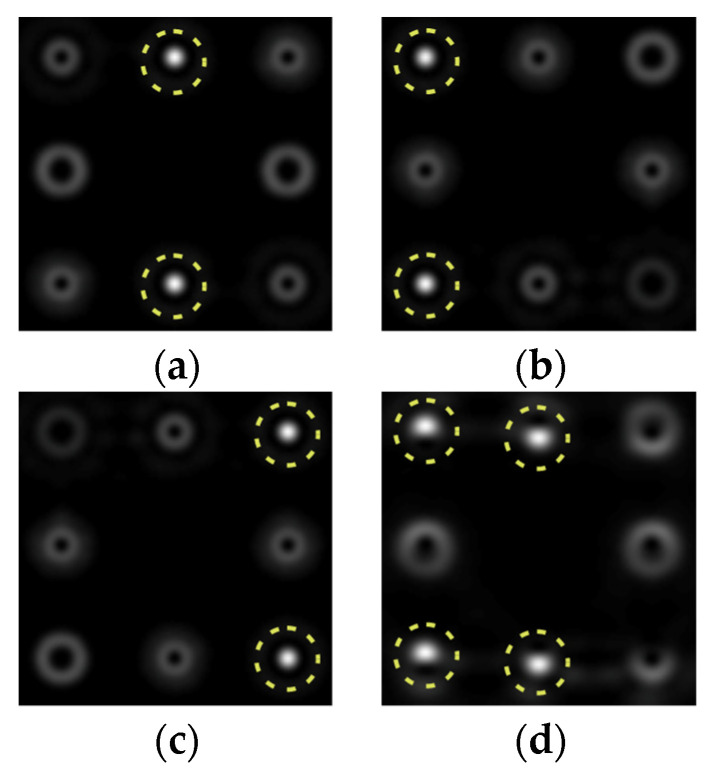
Action of an eight-channel DOE, matched with optical vortices in the analysis of vortex fields with cylindrical polarization of the second order (*p* = 2): (**a**) *m* = 0, (**b**) *m* = 1, (**c**) *m* = −1, (**d**) *m* = 0, 1, and (**e**) *m* = −1, 1.

**Figure 5 sensors-22-00890-f005:**
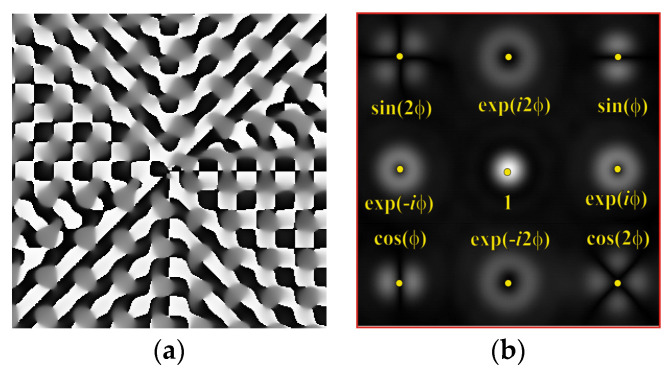
Nine-channel DOE, matched with optical vortices and their superpositions: (**a**) the form of the phase function, and (**b**) the correspondence of diffraction orders in the focal plane.

**Figure 6 sensors-22-00890-f006:**
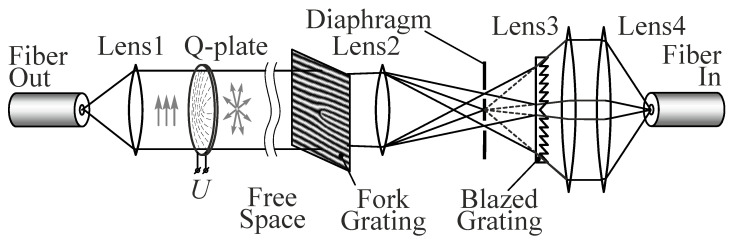
Schematic of an optical setup.

**Figure 7 sensors-22-00890-f007:**
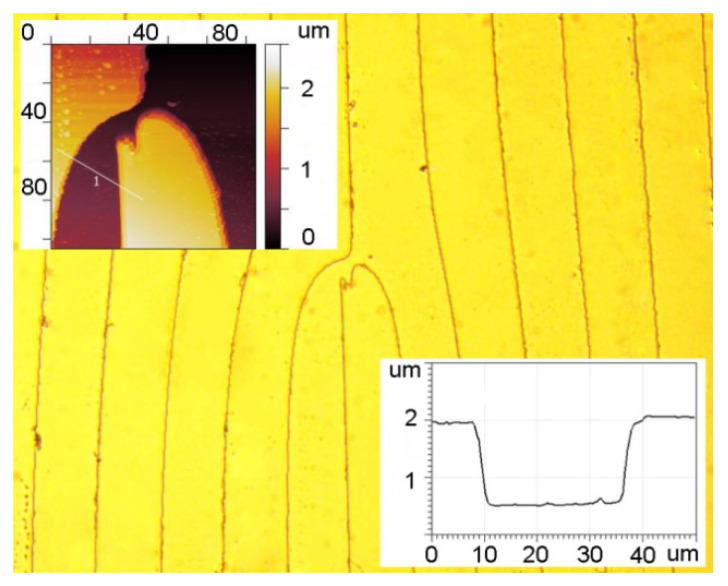
AFM images of the central segment and surface profile of a fork-shaped grating with a relief height of 1.7 μm and a period of 120 μm.

**Figure 8 sensors-22-00890-f008:**
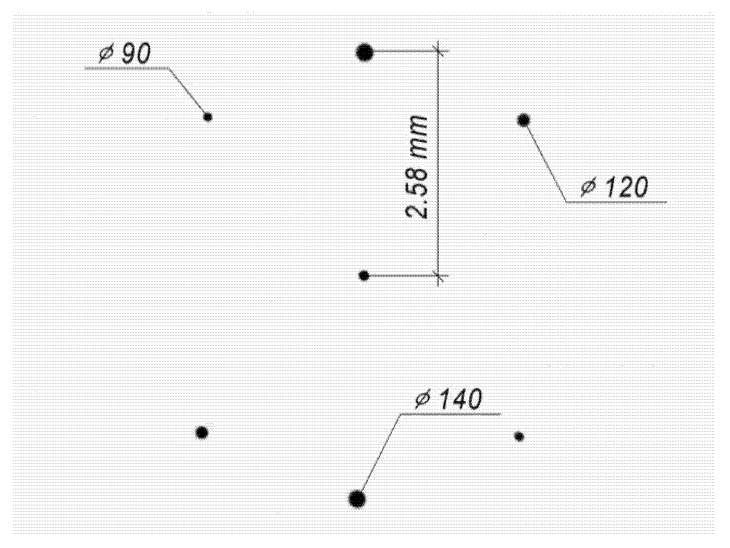
Image of the topology of point diaphragms.

**Figure 9 sensors-22-00890-f009:**
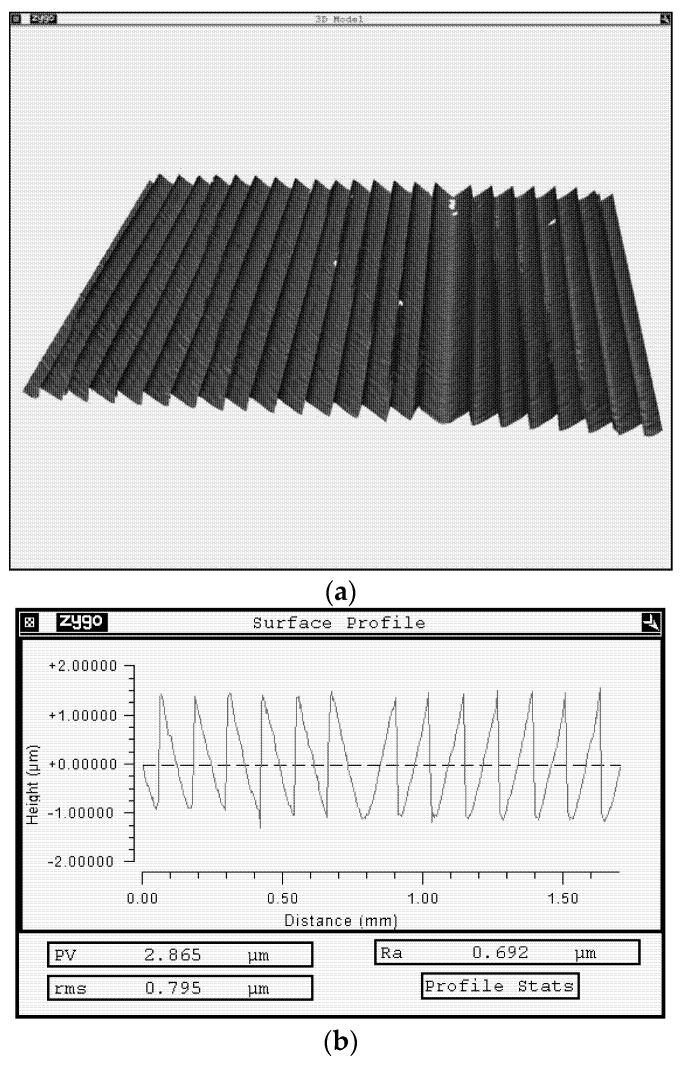
Image of the manufactured blazed grating: (**a**) 3D image of the relief, and (**b**) cross-section of the relief profile.

**Figure 10 sensors-22-00890-f010:**
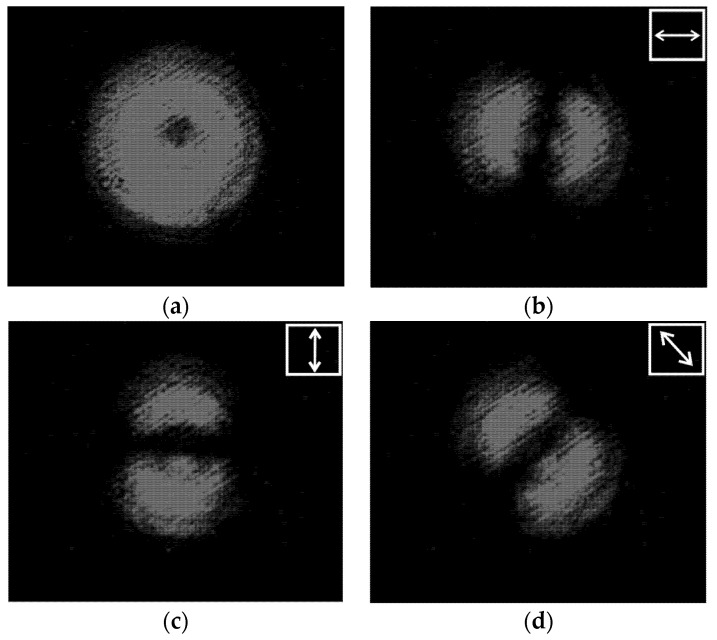
Intensity distributions of a radially polarized beam: (**a**) without a polarizer, and (**b**–**d**) with a polarizer for the axis inclined at (**b**) 0°, (**c**) 45°, and (**d**) 90°.

**Figure 11 sensors-22-00890-f011:**
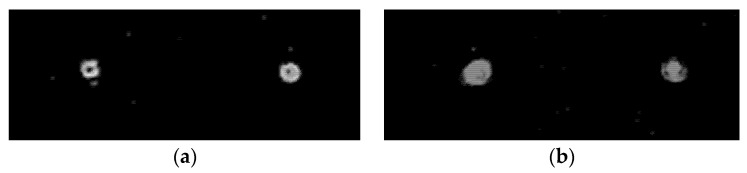
Intensity distributions in the focal plane of lens L2: (**a**) linearly polarized beam, and (**b**) azimuthally polarized beam.

**Figure 12 sensors-22-00890-f012:**
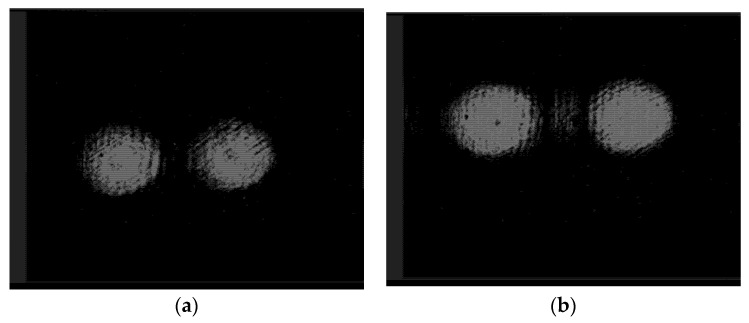
Intensity distributions after passing the blazed grating at a distance of (**a**) 60 mm and (**b**) 75 mm.

**Figure 13 sensors-22-00890-f013:**
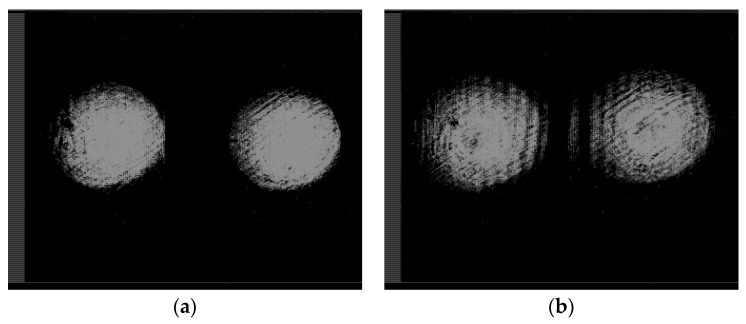
Intensity distributions in the diffraction orders after passing through lens L3 at a distance of (**a**) 50 mm and (**b**) 100 mm.

**Table 1 sensors-22-00890-t001:** Code correspondence of diffraction order numbers to the presence of phase vortices in the analyzed beam for a homogeneous polarization state.

Present Vortices	Code Correspondence							
	1	2	3	4	5	6	7	8
*m* = 0	1	0	0	0	1	0	0	0
*m* = 1	0	1	0	0	0	0	0	0
*m* = 2	0	0	1	0	0	0	0	0
*m* = 3	0	0	0	1	0	0	0	0
*m* = −1	0	0	0	0	0	1	0	0
*m* = −2	0	0	0	0	0	0	1	0
*m* = −3	0	0	0	0	0	0	0	1
*m* = −1, 1	0	1	0	0	0	1	0	0
*m* = 0, −1, 1	1	1	0	0	1	1	0	0
*m* = 1, −2	0	1	0	0	0	0	1	0
*m* = −1, 2, −3	0	0	1	0	0	1	0	1

**Table 2 sensors-22-00890-t002:** Code correspondence of diffraction order numbers to the presence of phase vortices in the analyzed beam for a cylindrical state of the first order polarization (*p* = 1).

Present Vortices	Code Correspondence							
	1	2	3	4	5	6	7	8
*m* = 2	0	1	0	1	0	0	0	0
*m* = −1	1	0	0	0	1	0	1	0
*m* = −2	0	0	0	0	0	1	0	1
*m* = 0, 1	1	1	1	0	1	1	0	0
*m* = 0, 2	0	1	0	1	0	1	0	0
*m* = −1, 1	1	0	1	0	1	0	1	0
*m* = 0, −1, 1	1	1	1	0	1	1	1	0
*m* = 1, −2	1	0	1	0	1	1	0	1
*m*= 0, −1, 2	1	1	0	1	1	1	1	0

**Table 3 sensors-22-00890-t003:** Code correspondence of diffraction order numbers to the presence of phase vortices in the analyzed beam for a cylindrical state of the second order polarization (*p* = 2).

Present Vortices	Code Correspondence							
	1	2	3	4	5	6	7	8
*m* = 0	0	0	1	0	0	0	1	0
*m* = 1	0	0	0	1	0	1	0	0
*m* = −1	0	1	0	0	0	0	0	1
*m* = 0, 1	0	0	1	1	0	1	1	0
*m* = 0, −1	0	1	1	0	0	0	1	1
*m* = 1, −1	0	1	0	1	0	1	0	1
*m* = 0, 1, −1	0	1	1	1	0	1	1	1

**Table 4 sensors-22-00890-t004:** Field distribution in the focal plane for beams with different polarizations when using a filter with vortex orders.

Beam Type, ${\left|G(u,v)\right|}^{2}$	${\left|G_{x}(u,v)\right|}^{2}$ $,\;\arg\left[G_{x}(u,v)\right]$	${\left|G_{y}(u,v)\right|}^{2}$ $,\;\arg\left[G_{y}(u,v)\right]$
Diagonal Linear, m = 1 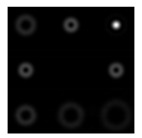	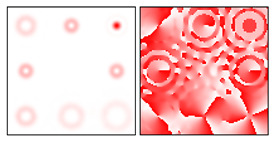	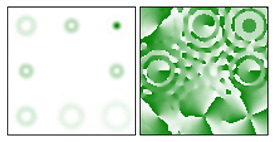
Right Circular, m = 1 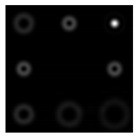	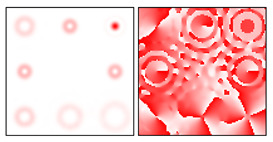	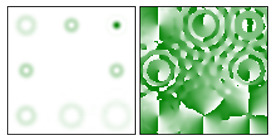
Radial 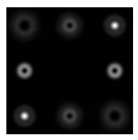	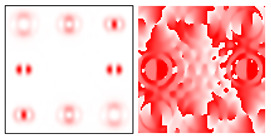	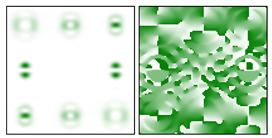
Azimuthal 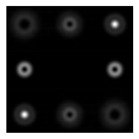	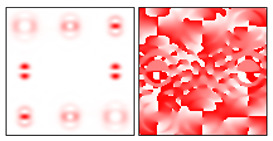	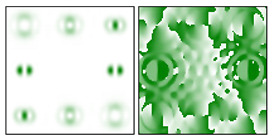

**Table 5 sensors-22-00890-t005:** Field distribution in the focal plane for cylindrically polarized beams of the first order (*p* = 1) when using a filter with a superposition of optical vortices.

$\mathrm{Beam}\;\mathrm{Type},\;{\left|G(u,v)\right|}^{2}$	${\left|G_{x}(u,v)\right|}^{2}$	${\left|G_{y}(u,v)\right|}^{2}$
Radial (*m* = 0) 	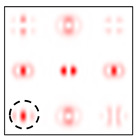	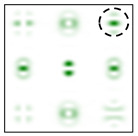
Azimuthal (*m* = 0) 	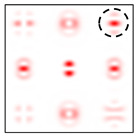	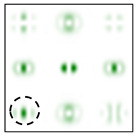
Radial (*m* = 1) 	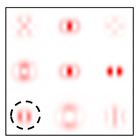	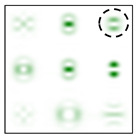
Azimuthal (*m* = 1) 	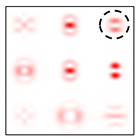	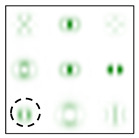

**Table 6 sensors-22-00890-t006:** Field distribution in the focal plane for cylindrically polarized beams of the second order (*p* = 2) when using a filter with a superposition of optical vortices.

$\mathrm{Beam}\;\mathrm{Type},\;{\left|G(u,v)\right|}^{2}$	${\left|G_{x}(u,v)\right|}^{2}$	${\left|G_{y}(u,v)\right|}^{2}$
Radial (*m* = 0) 	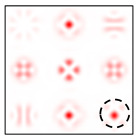	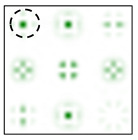
Azimuthal (*m* = 0) 	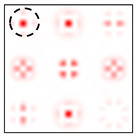	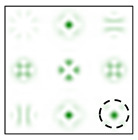
Radial (*m* = 1) 	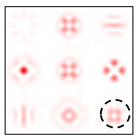	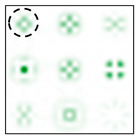
Azimuthal (*m* = 1) 	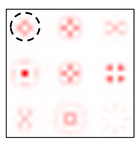	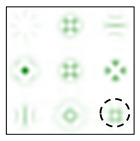

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
