# Peer review of "Free-Space Transmission and Detection of Variously Polarized Near-IR Beams Using Standard Communication Systems with Embedded Singular Phase Structures"

_sensors, 2022, doi:10.3390/s22030890_

Round 1

Reviewer 1 Report

This article theoretically analyzed the interaction of vortex beams of various orders with the main polarization state and the passage of beams with different polarization states through multi-order diffractive optical elements (DOEs) is simulated numerically, constructing tables of code correspondence of diffraction order numbers to the presence of phase vortices. I recommend it to be published after proper revision.

  • The interactions between vortex beams of different orders and the main polarization state and the theoretical description of the method for detecting the polarization state based on the superposition of orders are discussed in Chapter 2. It is recommended to summarize and analyze the above models at the end of this section.
  • The performed experiment studies the recognition of the first order cylindrical polarization state formed by a Q-plate converter using a phase DOE, which is not so complicated compared to the models in Chapter 2. Which systems do the different models analyzed in Chapter 2 mainly exist? Please elaborate on the necessity of these models.
  • The format of references is not uniform, such as abbreviations of magazine names, bolding, etc.
  • The article has some formatting issues, such as Fig. and Figure, and section title of Chapter 2. The author should make a full revision.

Author Response

The authors are grateful to the reviewers and the editor for the opportunity to make changes. We have submitted changes and responses to the comments made.

This article theoretically analyzed the interaction of vortex beams of various orders with the main polarization state and the passage of beams with different polarization states through multi-order diffractive optical elements (DOEs) is simulated numerically, constructing tables of code correspondence of diffraction order numbers to the presence of phase vortices. I recommend it to be published after proper revision.

  • The interactions between vortex beams of different orders and the main polarization state and the theoretical description of the method for detecting the polarization state based on the superposition of orders are discussed in Chapter 2. It is recommended to summarize and analyze the above models at the end of this section.

Reply. We have added possible applications in Chapter 2.

  • The performed experiment studies the recognition of the first order cylindrical polarization state formed by a Q-plate converter using a phase DOE, which is not so complicated compared to the models in Chapter 2.

Reply. In the experiment, we used a special case of the model, with the main attention being paid to the practical implementation of the introduction of a demultiplexed signal into the existing communication system using multi-order DOEs. At the same time, a high efficiency of the system for detecting signals encoded by the states of cylindrical polarization is critically important. To solve this problem, it was proposed to use an optical system consisting of a multi-order DOE, lenses, a diaphragm located in the focal plane, and a system for efficient signal coupling into a fiber-optic detector, consisting of a bidirectional balzed grating, and collimating lenses.

  • Which systems do the different models analyzed in Chapter 2 mainly exist? Please elaborate on the necessity of these models.

Reply. The approaches using the described models make it possible to expand the capabilities of already existing multichannel systems [17]. The description is added at the end of section 2.

  • The format of references is not uniform, such as abbreviations of magazine names, bolding, etc.

Reply. Fixed.

  • The article has some formatting issues, such as Fig. and Figure, and section title of Chapter 2. The author should make a full revision.

                Reply. Fixed.

Reviewer 2 Report

The manuscript entitled “Free-space transmission and detection of variously polarized near-IR beams using standard communication systems with embedded singular phase structures” presents a good theoretical and experimental research work on the transmission of information in free space through several channels using the polarization state.  

However, there are a list of points that need improvement before being considered for publication:

The introduction section is too big. It contains too much information that should be reduced or pointed to external references. Other information can be theoretical results so should be in a different section.

Figure 3-a and 3-b needs a spacer.

The section 3. Results of the experiment, is poor. A set of figures are presented but a strong discussion is missing.

The figures must transmit all the information it-self’s so some legends must be added and improved.

Figure 11 is too small, it’s hard to distinguish both intensity distributions.

Author Response

The authors are grateful to the reviewers and the editor for the opportunity to make changes. We have submitted changes and responses to the comments made.

 The manuscript entitled “Free-space transmission and detection of variously polarized near-IR beams using standard communication systems with embedded singular phase structures” presents a good theoretical and experimental research work on the transmission of information in free space through several channels using the polarization state.  

However, there are a list of points that need improvement before being considered for publication:

The introduction section is too big. It contains too much information that should be reduced or pointed to external references. Other information can be theoretical results so should be in a different section.

Reply. In terms of structure and volume, the introduction corresponds to a full-length article and is aimed at a wide readership. By tradition, the section may contain the main theoretical provisions. A detailed consideration of theoretical information allows one to have a complete understanding of the proposed method. The "Article" format assumes this kind of detailed presentation.

Figure 3-a and 3-b needs a spacer.

Reply. Fixed.

The section 3. Results of the experiment, is poor. A set of figures are presented but a strong discussion is missing.

The figures must transmit all the information it-self’s so some legends must be added and improved.

Reply. An explanation has been added to the text of the manuscript.

Figure 11 is too small, it’s hard to distinguish both intensity distributions.

Reply. The scale of Fig. 11 is enlarged, but the resolution of the infrared camera used is not enough to show in detail the presence of peaks and rings in the intensity distribution. Unfortunately, the scale of images in the frequency plane is determined by the wavelength and focal length of the Fourier lens and cannot be changed. We have added explanations to the figure describing what is in the image. Description of the camera with which the images were taken has been added.

Reviewer 3 Report

The authors proposed to achieve multichannel information transmission in free space by means of variously polarized beams. The results are interesting. This manuscript can be published after the following revisions.

  1. What the meaning of the coordinate eh in Eq.1.
  2. The overall contribution and motivation of this paper are also not elaborately discussed in the introduction. This part requires substantial improvement.
  3. The authors should discuss the intensity distribution of the Eq.3.
  4. In Fig. 2, the Figs.(a)-(e) are not separated. There is not the zero-order diffraction beam in Figs.2-4 and table 4, the authors should explain it.
  5. What is the meaning of the coordinate ep in Eq.7.
  6. The authors should check the Fig.3, the mark is not match for the intensity distribution of the diffraction beam.
  7. Fig.11b is too fuzzy, and the maximum value of the center of the series is not easy to distinguish.

Author Response

The authors are grateful to the reviewers and the editor for the opportunity to make changes. We have submitted changes and responses to the comments made.

The authors proposed to achieve multichannel information transmission in free space by means of variously polarized beams. The results are interesting. This manuscript can be published after the following revisions.

  1. What the meaning of the coordinate eh in Eq.1.

Reply. We have added an explanation to the text.

The subscript h indicates the uniform polarization state vector.

  1. The overall contribution and motivation of this paper are also not elaborately discussed in the introduction. This part requires substantial improvement.

Reply. We have added the description in the Introduction

.

  1. The authors should discuss the intensity distribution of the Eq.3.

Equation 3 corresponds to the complex amplitude of the transmission of one of the considered spatial filters. Figure 3 shows the intensity distributions in the focal plane when the filter from equation 3 is illuminated with different beams. Details are indicated in the caption below. Figure 3 has been revised.

  1. In Fig. 2, the Figs.(a)-(e) are not separated. There is not the zero-order diffraction beam in Figs.2-4 and table 4, the authors should explain it.

Reply. The zero order (m = 0) is shifted from the optical axis in accordance with formula 3 by adding the carrier frequency. The central part of the focal plane is not used for useful information to reduce the influence of interferences localized in the central part.

  1. What is the meaning of the coordinate ep in Eq.7.

Добавлено

Reply.  is the vector of the cylindrical polarization of the pth order.

  1. The authors should check the Fig.3, the mark is not match for the intensity distribution of the diffraction beam.

Reply. Fixed.

  1. Fig.11b is too fuzzy, and the maximum value of the center of the series is not easy to distinguish.

Reply. The scale of Fig. 11 is enlarged, but the resolution of the infrared camera used is not enough to show in detail the presence of peaks and rings in the intensity distribution. Unfortunately, the scale of images in the frequency plane is determined by the wavelength and focal length of the Fourier lens and cannot be changed. We have added explanations to the figure describing what is in the image. Description of the camera with which the images were taken has been added.

Round 2

Reviewer 1 Report

I appreciate the authors for the detailed reply. The authors have addressed all the concerns, and this paper can be accepted as it is.